# Integrated Cardiorespiratory Rehabilitation and Its Impact on Cardio–Renal–Metabolic Profile After Cardiac Surgery

**DOI:** 10.3390/nu16213699

**Published:** 2024-10-30

**Authors:** Stefanie Marek-Iannucci, Alberto Palazzuoli, Matteo Babarto, Zlatan Lazarevic, Matteo Beltrami, Francesco Fedele

**Affiliations:** 1Department of Cardiovascular Rehabilitation, San Raffaele, Monte Compatri, 00040 Rome, Italy; stefanie.marekiannucci@gmail.com (S.M.-I.); zlatan.lazarevic@sanraffaele.it (Z.L.); 2Cardiovascular Diseases Unit, Cardio-Thoracic and Vascular Department, Le Scotte Hospital University of Siena, 53100 Siena, Italy; palazzuoli2@unisi.it; 3Department of Arrhythmia and Electrophysiology, Careggi University Hospital, 50134 Florence, Italy; 4IRCCS San Raffaele Cassino, 03043 Cassino, Italy

**Keywords:** lifestyle, rehabilitation, cardiovascular diseases, risk factors, aerobic exercise

## Abstract

Background: Cardiovascular diseases (CVDs) and chronic kidney disease (CKD) are common causes of morbidity and mortality. However, the impact of changes in lifestyle and rehabilitation programs on the progression of cardiovascular, renal, and metabolic (CRM) conditions, remains unclear. Methods: In a retrospective manner, we analyzed charts of 200 patients admitted for cardiorespiratory rehabilitation at our facility in 2023. A 6 min walk test, echocardiographic features, and laboratory values were investigated to evaluate the impact of cardiorespiratory rehabilitation in patients post cardiac surgery. This study examined the impact of combined lifestyle and exercise scores (diet, alcohol consumption, smoking, aerobic physical activity, sedentary behavior, sleep duration, and social connection) on cardio–renal–metabolic profiles and on a quality-of-life score measured by the Borg Scale. Results: During the rehabilitation program, left ventricular ejection fraction (LVEF) significantly increased (51.2 vs. 54.3%, SEM 0.51 *p* = 0.001). The six-minute walk test (6 MWT) significantly improved in terms of meters (133 vs. 373 m, SEM 6.41, *p* < 0.001) and Borg scale (6.6 vs. 2.5, SEM 0.06, *p* < 0.001). Glycemia levels reduced significantly (114.5± vs. 107.4± mg/dL, SEM 2.45, *p* = 0.001). While total cholesterol levels (119.4 vs. 129.6 mg/dL, SEM 2.4, *p* < 0.001) as well as HDL levels (29.9 vs. 40 mg/dL, SEM 0.62, *p* < 0.001) significantly increased, triglyceride levels significantly decreased (128.5 vs. 122.1 mg/dL, SEM 3.8, *p* = 0.048). There was no change in LDL levels. Creatinine levels remained stable throughout the period of rehabilitation. Conclusions: Cardiorespiratory rehabilitation has a significant impact on myocardial function, quality of life in terms of exercise capacity and symptoms (6 MWT) as well as laboratory levels relevant for cardiovascular prevention such as glycemia and lipid profile.

## 1. Introduction

Cardiovascular and renal complications are leading causes of mortality and morbidity in patients with coronary artery disease (CAD), with cardiovascular disease (CVD) contributing to up to one-third of deaths [1,2,3,4,5,6]. Sedentary lifestyle and obesity-associated cardiometabolic disorders including hypertension, diabetes, and cardiovascular disease share common pathophysiological features and often cluster in affected individuals. The metabolic–vascular link among people living with obesity and/or type 2 diabetes is associated with higher rates of both CVD and chronic kidney disease (CKD) [7]. Additionally, CAD is often associated with more severe atherosclerosis in other districts and renal dysfunction. The contemporary presence of risk factors such as smoking, hypertension, and diabetes leads to glomerulosclerosis and CKD. Previous data evidenced a correlation between cardiovascular, renal, and metabolic (CRM) conditions, including hemodynamic changes, neurohormonal imbalances, mitochondrial superoxide production, and exacerbation of oxidative stress [3,6]. Given the fatal outcomes and the strong interplay between risk factors, lifestyle, CVD, and CKD, the progression of coexisting CRM conditions has been of interest to many scientific groups [2,3,6,8,9]. Research on disease progression covers the natural trend from health to morbidity and furthermore from multimorbidity to death, providing more detailed information, essential for primary and secondary prevention of disease as well as patient-specific management. Pharmacologic advances and medical interventions aim to target risk factors in patients with established cardiometabolic disease, providing cardio–renal protection. However, data are still scarce regarding the progression of CRM conditions, and effective methods to counteract the latter are limited. Changes in lifestyle are considered easy interventions to prevent or delay disease progression. The associations between certain rehabilitation programs, including lifestyle factors, diet habits, as well as physical activity, and their effects on cardiovascular (CV) and renal outcomes are not extensively investigated [10,11,12]. While previous studies have researched the impact of lifestyle on the onset of CVD [10], only a few studies estimate the relationships between rehabilitation plans, lifestyle factors, and dynamic disease progression.

Early commencement of aerobic exercise after cardiac surgery significantly improves functional and aerobic capacity. Cardiac rehabilitation is a complex intervention that requires a multidisciplinary team, including health and physical activity promotion, psychological support, and cardiovascular risk management, tailored for each individual. Despite the well-established beneficial effects of cardiac rehabilitation after cardiac surgery, this intervention strategy still remains underutilized due to low rates of referrals, adherence, and compliance.

Furthermore, previous studies have neglected emerging lifestyle factors (such as Mediterranean diet, sedentary behavior, and social connection) and have not taken into account the relative contribution of each factor. This could be of the essence regarding the development of effective lifestyle interventions and help construct alternative treatment plans related to the progression of CVD.

To address the above-mentioned research gaps, this study focused on the effect of cardiorespiratory rehabilitation in the immediate post-surgical period on physical capacity and general health status as measured by laboratory values relevant to cardiovascular risk quantification.

## 2. Methods

A total of 200 patient charts were analyzed in a retrospective manner in our rehabilitation facility. All patients admitted to the cardiorespiratory rehabilitation center San Raffaele, Monte Compatri, Lazio, Italy, in 2023 were eligible for inclusion in this study. Exclusion criteria were incomplete charts, transfer back to the originating hospital for urgent medical intervention, or discharge against medical advice. In a retrospective manner, we collected echocardiographic data (left ventricular ejection fraction) and laboratory markers relevant to the development of cardiovascular disease (total cholesterol, LDL, HDL, triglycerides, glucose levels, creatinine) at the time of admission and after completion of the program.

Dietary regimen followed an individual patient-based plan, based on the Mediterranean diet with focus on low sodium and fat intake, adequate calories, and balanced carbohydrate, protein, and vegetable intake. Briefly, a Mediterranean diet consists of a balanced intake of a minimum of four out of seven food groups, which are considered dietary priorities: including fruits, vegetables, fish, processed meats, unprocessed red meats, whole grains, and refined grains.

The physical rehabilitation program was as follows: On day one post admission, patients underwent a 6 min walk test (6 MWT) and were started on a reconditioning program on either a treadmill or bicycle. Depending on the general state of the patient, they would be allocated additionally to a respiratory gymnastic group training. For the following 20 days, patients accessed the physical rehabilitation facilities 2–3 times per day where they alternated between phases of muscular reconditioning (treadmill, bicycle) or respiratory exercises. After this period, the patients underwent ergometry testing, leading to a personalized physical activity program depending on the patient’s fitness as prescribed by the attending physician. The day prior to discharge, the patients underwent another 6 MWT to assess for physical improvement.

For statistical analysis, we used GraphPad Prism 10. Paired *t*-test was used to analyze the data pre and post rehabilitation for each patient. For subgroup analysis between coronary artery bypass graft (CABG) and valve surgery patients, 2-way-ANOVA analysis was used. Results were considered statistically significant when the *p*-value was inferior to 0.05.

The patients included in this study followed an extensive rehabilitation protocol including regular physical activity, diet modification, and risk factor control. As per previously published data [2,13], the diagnosis of CV events and CKD were attributed as recommended in the International Classification of Diseases.

### Assessment of Lifestyle Factors and Other Covariates

Regular physical activity was defined as ≥150 min moderate activity per week, ≥75 min vigorous activity per week, an equivalent combination, moderate activity ≥ 5 times/week, or vigorous activity ≥ 1 time/week. Low-to-moderate sedentary behavior (≤4 h per day watching TV and using a computer) was classified as low risk. Low-risk social connection was defined as frequent social connection (≤1 on the social isolation index). Social isolation index was calculated based on the sum of the following three indices: the number in the household, frequency of friend/family visits, and participation in leisure/social activity [14].

## 3. Results

### 3.1. Patients Characteristics

The mean age and standard deviation of the patient population within this study was 66.5 ± 10.5 years. A total of 72.5% of patients were male and 27.5% female. A total of 60.5% of the patients underwent CABG, 30.5% underwent single or multipole valve replacement or repair, and 9% underwent a combined CABG and valve surgery or other procedures, such as pacemaker implantation, atrial tumor resection or pericardiectomy, prior to admission to rehabilitation. Regarding cardiovascular comorbidities, 33% of patients had diabetes mellitus type 2, 77.5% arterial hypertension, and 66% dyslipidemia (Table 1).

There was a significant increase (51.2 vs. 54.3%, SEM 0.51, *p* = 0.001) in left ventricular ejection fraction (LVEF) between admission and discharge from rehabilitation (Figure 1). The six-minute walk test significantly improved from the time of admission to rehabilitation compared to after completion of the program in terms of meters (133 vs. 373 m, SEM 6.41, *p* < 0.001) and Borg scale (6.6 vs. 2.5, SEM 0.06, *p* < 0.001), as shown in Figure 2A and 2B, respectively. Glycemia levels at the time of admission were significantly higher (114.5± vs. 107.4± mg/dL, SEM 2.45, *p* < 0.001) than after completion of rehabilitation. Interestingly, total cholesterol levels (119.4 vs. 129.6 mg/dL, SEM 2.4, *p* < 0.001) as well as HDL levels (29.9 vs. 40 mg/dL, SEM 0.62, *p* < 0.001) significantly increased from the time of admission to discharge of the rehabilitation facility. Triglyceride levels significantly decreased (128.5 vs. 122.1 mg/dL, SEM 3.8, *p* = 0.048) from discharge to completion of the rehabilitation program, while there was no change in LDL levels. Regarding kidney function, creatinine levels remained stable throughout the period of rehabilitation (Table 2).

### 3.2. Cardiovascular Rehabilitation in Patients with CABG Intervention Compared to Valve Surgery

Subgroup analysis showed that LVEF recovery between pre and post cardiovascular rehabilitation is significantly higher (*p* < 0.05) in the CABG compared to the valve surgery group (Figure 3A). Regarding the 6 MWT, both patient populations (CABG and valve surgery) had significant improvement (*p* < 0.0001) in symptoms and functional capacity, when comparing pre to post rehabilitation, but did not differ between the groups (Figure 3B,C).

It is noteworthy that total cholesterol and LDL levels were lower in the CABG group when compared to the valve surgery group, as recommended by the guidelines for cardiovascular disease prevention [1] (Figure 4A,B). Both CABG and valve surgery groups showed a significant increase (*p* < 0.001) in HDL throughout the rehabilitation program, likely due to the lifestyle changes throughout the rehabilitation program (Figure 4C), resulting in a total cholesterol increase, which reached statistical significance (*p* < 0.01) only in the CABG group.

### 3.3. The Role of Combined Exercise Activity Program and Diet Modification on Quality of Life and Outcome

Our data showed a significant improvement in LVEF as well as a significant increase in the 6 MWT performance in meters throughout the rehabilitation period. Additionally, there was a significant reduction in symptoms according to the Borg scale during the 6 MWT when comparing the time of admission to the rehabilitation facility and discharge. This indicates that cardiorespiratory rehabilitation after cardiac surgery significantly impacts the quality of life (QoL) of the patient in terms of physical capacity improvement and symptom reduction.

## 4. Discussion

This study evaluated the role of lifestyle factor modifications with an integrated diet and rehabilitation program on the progression of CRM conditions in a consecutive group of patients after cardiac surgery. Our findings suggested that patients who completed cardiorespiratory rehabilitation experienced a lower CV risk burden associated with improvements in physical health and fitness level. The lifestyle modifications, including smoking interruption, physical activity, and diet modification with a specific KCal threshold, significantly modified the CMR risk. Similarly, diet and exercise training played an important role in specific stages of disease progression. Finally, our rehabilitation program was associated with a significant improvement in QoL, measured by a significant reduction of symptoms (Borg Scale), better exercise tolerance, significantly improved LVEF and exercise capacity measured by 6 MWT, and reduced events in terms of urgent visit or rehospitalization.

CVD, type 2 diabetes, and CKD significantly impact morbidity and mortality and are of great interest and concern to the AHA, the American Diabetes Association (ADA), and the European Society of Cardiology (ESC) [8,9,15]. However, little is known about the relationship between specific risk factor modification programs and the progression of CAD and CRM conditions. A cross-sectional study investigated the prevalence and overlap of CRM conditions, indicating a tendential increase in CRM [1]. Other studies have focused exclusively on multimorbidity in patients with a specific risk profile [10,16]. Strict control of risk factors associated with daily aerobic exercise training leads to better risk factor profiles and QoL conditions.

Other studies have focused on the mere association between lifestyle and specific disease states (i.e., the onset of a specific CRM condition) [11,17,18] or the mortality among patients with CRM conditions [10]. The present study found that lifestyle significantly impacts the dynamic progression of CRM conditions as soon as in the immediate post-surgical period. Our findings align with existing studies that have focused exclusively on a particular disease stage of a specific CRM condition [10,17,18]. More importantly, this study contributes to the existing evidence regarding the association of lifestyle behaviors with dynamic progression in terms of both the number and types of CRM conditions.

Our findings indicate that combined exercise and healthy lifestyle programs have had a protective effect in terms of both NYHA and CCS class improvement. Additionally, our results indicate a slightly stronger association with the development of type 2 diabetes compared to the transition to other CRM condition types. This is in accordance with previously published findings regarding cardiometabolic multimorbidity [19] as well as cancer-related multimorbidity and cardiometabolic diseases [20]. This might be due to the fact that healthy lifestyle behaviors (i.e., diet and exercise) are directly related to insulin sensitivity, a key factor in the development of type 2 diabetes [21,22]. During the rehabilitation period, diabetic patients were treated with long- and short-acting insulin based on a sliding scale methodology to ensure well-controlled glycemia levels. The latter is of the essence in order to prevent wound healing delays or infections. Patients with dyslipidemia were treated with a combination of a high-intensity statin and ezetimibe as per ESC guidelines [1].

Four conventional lifestyle factors have been studied in the past (diet, alcohol consumption, smoking, and physical activity), while emerging factors such as sedentary behavior, sleep duration, and exercise program often have been neglected [10,19,20,23]. A previous study [24] demonstrated a superior association between never smoking and disease progression, indicating that smoking may be an essential target to address. Smoking enhances chronic inflammation, endothelial dysfunction, and other CRM-related pathway abnormalities [25,26,27]. High seafood and polyunsaturated fatty acid intake might be an explanation for the significant association between diet and cardiometabolic multimorbidity, shown in previous publications [28]. The Mediterranean diet is based on traditional eating patterns providing emphasis on whole, minimally processed foods, including fruits, vegetables, whole grains, nuts, and seeds. High healthy fat consumption consists of olive oil, fish oil with limited intake of saturated fats and limited red meat. This habit is associated with high consumption of plant-based foods, including legumes, fruits, and vegetables, and low-fat cheese such as yogurt and fresh-derived milk products. This type of diet provides several potential advantages on the glycemic and lipid profile: The high fiber content from fruits, vegetables, and whole grains aids in regulating blood sugar levels, enhancing insulin sensitivity, and reducing the risk of type 2 diabetes. Low saturated fat intake can reduce LDL cholesterol levels and increase HDL cholesterol, improving overall lipid profiles.

Antioxidant-rich foods within the diet help reduce markers of inflammation, potentially lowering the risk of atherosclerosis and restoring endothelial function. The high vegetable and fruit content facilitates antioxidants and anti-inflammatory compounds, protecting against kidney disease progression. A moderate animal protein and salt intake reduces the workload on the kidneys which is beneficial for maintaining healthy blood pressure and vessel properties while reducing parietal stiffening and constriction.

Other studies evidenced an association between social isolation and a high risk of one CRM condition and mortality [29,30,31]. Furthermore, this study underlines the important impact of integrated rehabilitation on the progression of CRM conditions. In conclusion, a healthy lifestyle seems to have a protective effect on changes in CRM features. Smoking cessation, diet, and aerobic exercise significantly impacted specific disease transitions. Our study also suggests that lifestyle modifications associated with specific rehabilitation programs have preventive potential for the progression of CRM conditions, with QoL improvement already in the immediate post-surgical period.

## 5. Strengths and Limitations

The present study addressed, to our knowledge, for the first time, the relationship between lifestyle (including four conventional factors and three emerging factors) and the dynamic progression of CRM conditions using a purposed program. The contribution of each lifestyle factor was estimated to tackle rehabilitation and treatment strategies. In addition, the progression of CRM conditions and the role of combined lifestyle programs on CV risk burden and CAD disease evolution were assessed.

Nonetheless, limitations are worth noting. Patients received different antidiabetic and antihypertensive treatments which might be a bias, despite all patients being submitted to a similar program regardless of the therapy administered. We did not measure lifestyle habits and exercise habits at baseline and could not capture the possible lifestyle changes during follow-up. A longer observational period may be warranted to confirm the beneficial effects of our program on CV outcomes. The participants in this study were primarily Caucasians with a mean age of 66.5 ± 10.5 years. Caution is needed when generalizing our findings to other race and age populations.

The benefits of rehabilitative interventions are generally quantified as changes in VO2 with cardiopulmonary exercise tests. However, improvement in functional capacity is the most immediate and objective result of an effective cardiorespiratory rehabilitation program, which is easily assessable with a cost-efficient 6 MWT. Finally, despite accounting for confounders, the presence of residual confounding cannot be entirely ruled out.

## 6. Conclusions

In conclusion, the present study indicates that combined dietary and rehabilitation programs may produce a protective effect throughout the dynamic progression of CAD and CRM risk. An integrated approach including rehabilitation program lifestyle modification and dietary change leads to a better CV risk profile reduction and better QoL. Cardiac rehabilitation associated with a dietary program, plays a crucial role in managing metabolic diseases, by providing comprehensive support that addresses physical, psychological, and educational needs. Its multidisciplinary approach enhances physical health together with a holistic improvement in well-being. Current easily applicable programs demonstrated optimal learning education with patients self-monitoring their health metrics, promoting long-term adherence to healthy behaviors. The gradual physical conditioning, increasing overall fitness levels enhances the whole QoL while reducing atherosclerosis risk and slowing down physiologic kidney function decline.

## Figures and Tables

**Figure 1 nutrients-16-03699-f001:**
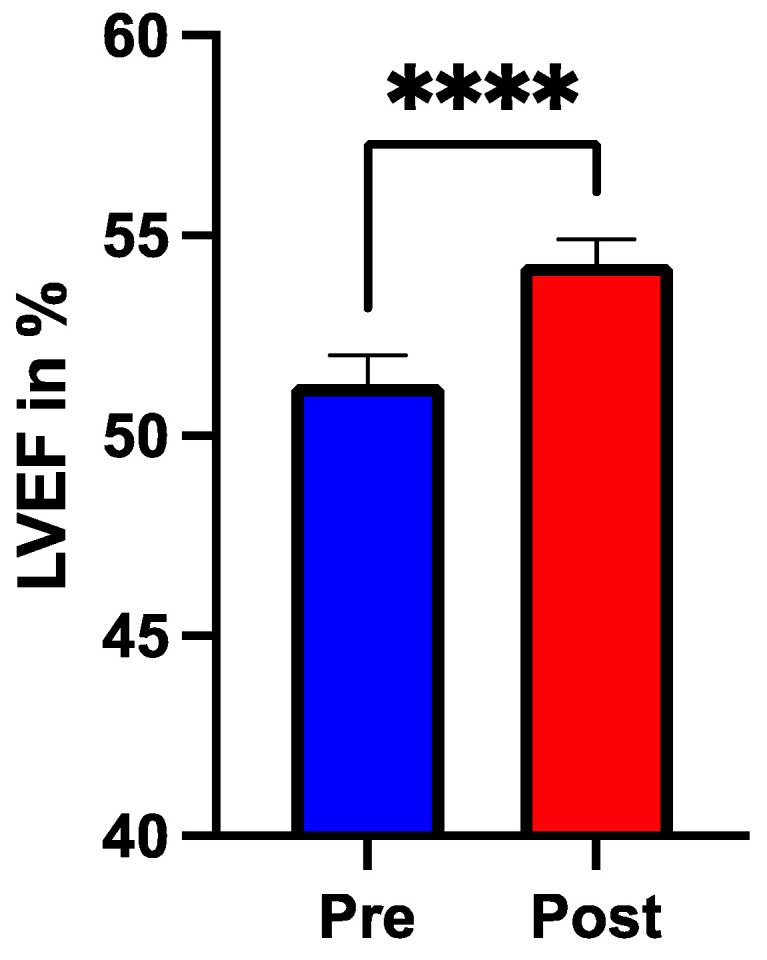
Left ventricular ejection fraction (LVEF) measured by echocardiography and quantified with the Simpson method before (blue bar “pre”) and after (red bar “post”) completion of rehabilitation. **** *p* < 0.0001.

**Figure 2 nutrients-16-03699-f002:**
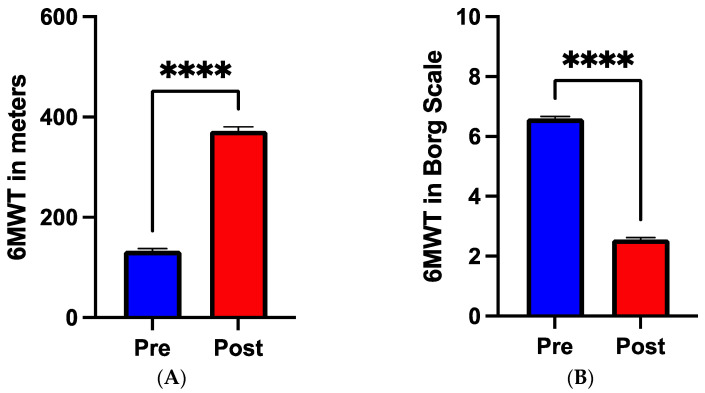
Results of the 6 min walk test (WT). (**A**) distance in meters performed at the WT before (blue bar) and after (red bar) completion of rehabilitation. (**B**) Severity of symptoms during exercise measured by Borg Scale before (blue bar) and after (red bar) completion of rehabilitation. **** *p* < 0.0001.

**Figure 3 nutrients-16-03699-f003:**
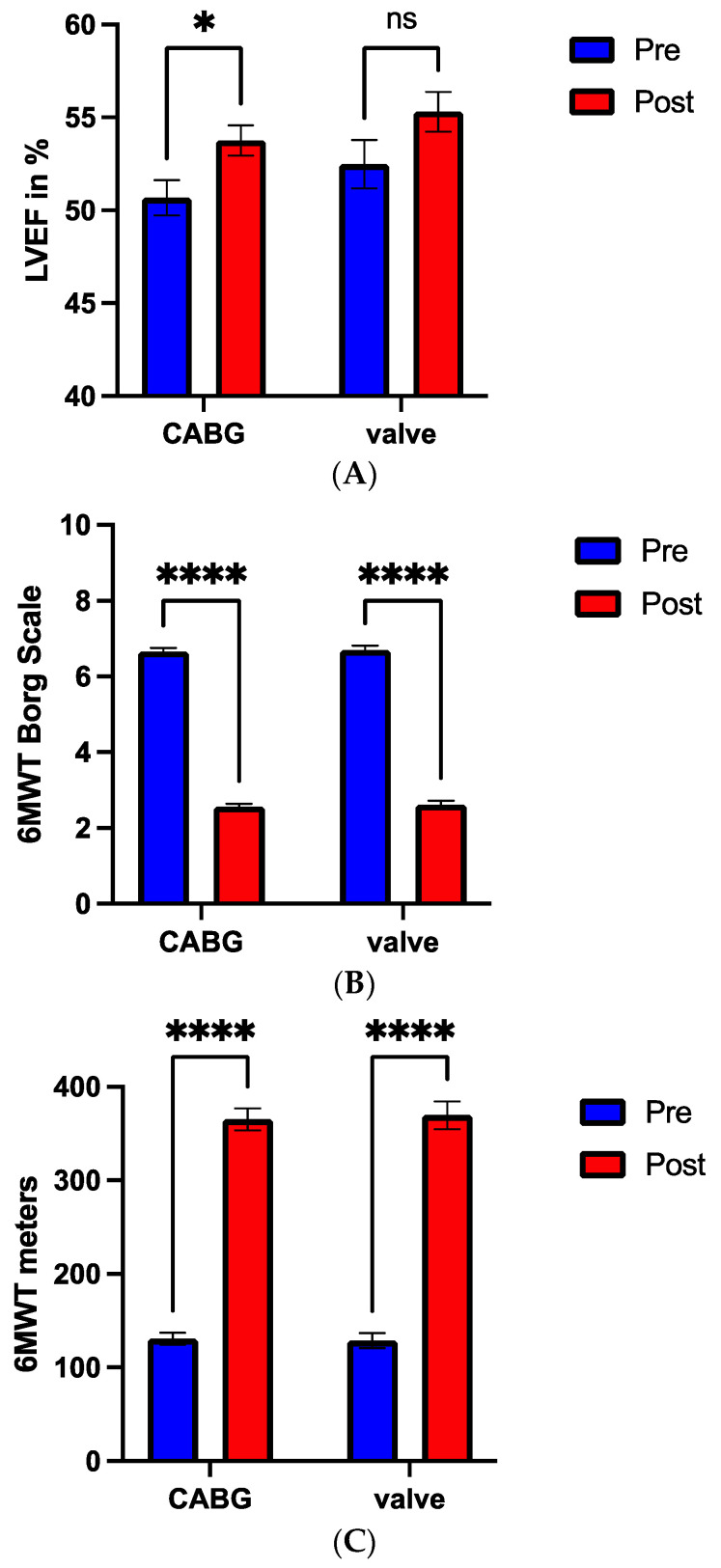
LVEF recovery (**A**), 6 MWT Borg scale (**B**), and 6 MWT meters (**C**) between pre (blue bar) and post (red bar) cardiovascular rehabilitation in CABG compared to valve surgery group. * *p* < 0.05, **** *p* < 0.0001, ns: not significant.

**Figure 4 nutrients-16-03699-f004:**
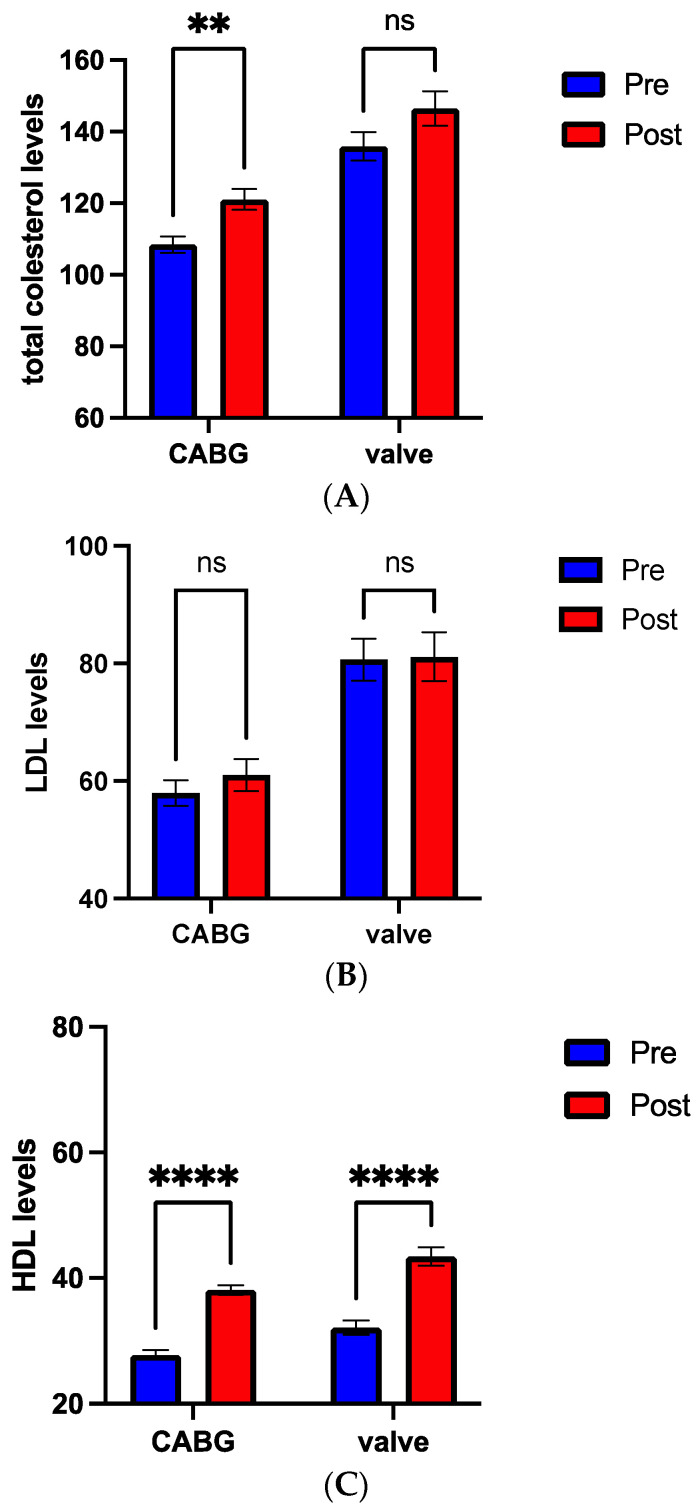
Total cholesterol (**A**), LDL levels (**B**), and HDL levels (**C**) between pre (blue bar) and post (red bar) cardiovascular rehabilitation in coronary artery bypass graft (CABG) compared to valve surgery group. ** *p* < 0.05, **** *p* < 0.0001, ns: not significant.

**Table 1 nutrients-16-03699-t001:** Patient characteristics. SD standard deviation, CABG coronary artery graft bypass, HTN arterial hypertension, DM2 diabetes mellitus type 2.

Mean Age and SD	66.5 (±10.5)
Gender	
Male	72.5%
Female	27.5%
Surgical procedure	
CABG	60.5%
Valve repair	30.5%
Combined or other surgery	9%
Comorbidities	
DM2	33%
HTN	77.5%
Dyslipidemia	66%

**Table 2 nutrients-16-03699-t002:** Significant changes in laboratory values before and after completion of rehabilitation. ns: not significant.

	Admission	Discharge	*p*-Value
Glycemia	114.52	107.41	<0.001
Total cholesterol	119.43	129.6	<0.001
HDL	29.94	40.01	<0.001
Triglycerides	128.57	122.12	<0.05
LDL	66.81	67.85	ns
creatinine	1.05	1.07	ns

## Data Availability

The original contributions presented in the study are included in the article, further inquiries can be directed to Stefanie Marek-Iannucci.

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
