# Peer review of "Integrated Cardiorespiratory Rehabilitation and Its Impact on Cardio–Renal–Metabolic Profile After Cardiac Surgery"

_nutrients, 2024, doi:10.3390/nu16213699_

Round 1
Reviewer 1 Report
Comments and Suggestions for Authors
I find that the study adresses an important issue which, unfortunately is sometimes overlooked. However, I would like to highlight some issues which may warrant further attention.
1. The title does not reflect the study of renal and metabolic issues which are mentioned in the mnuscript, therefore these should be added to better reflect the subject.
2. The Introducgtion could be expanded with further information on the cardiac, renal and metabolic issues.
3. Also, the cardiopulmonary exercise testing could be discussed as a mean of determining the functional capacity of patients.
4. It would be interesting to present the results also based on their preoperatory sedentary behaviour and based on the type of surgical intervention, since you included patients who suffered CABG, PCI or valve repair and the results may differ.
5. Were the patients under therapy for diebetes or hypercholesterolemia? If so, please expand the discussions.
6. Please expand the discussions based on the diet prescribed to your patients and the diets currently recommended (e.g. Mediteranean, DASH).
7. The benefits of cardiac rehabilitation are well-known, so perhaps you should address the unique findings of your particular study as compared to others in the conclusions.
Comments on the Quality of English Language
Please revise the English language as there are several mistakes and some sentences could be rephrased for better understanding.
Reviewer 2 Report
Comments and Suggestions for Authors
Reviewers' Comments to Authors:
The study addresses an important topic by conducting a scoping review on integrated cardio-respiratory rehabilitation and its impact on patients following cardiac surgery. While this is an interesting and relevant study, the authors should consider the following suggestions:
Abstract:
- Please include units in brackets, e.g., “The six-minute walk test significantly improved in terms of meters (133 versus 373 meters, p<0.001) [median/mean].”
Introduction:
- The introduction is well-written; however, the authors have not elaborated on cardio-respiratory rehabilitation and its impact on patients post-cardiac surgery. Please emphasize the study's relevance and significance, particularly regarding the immediate post-operative period, as outlined in the study's objective.
- Define abbreviations upon their first appearance, e.g., CKD
- Please clarify the term "CV."
Methods:
- There are several methodological concerns. The methods section is underdeveloped. The authors have not outlined the inclusion criteria, and there is a lack of information on key variables mentioned in the abstract (e.g., laboratory values relevant for cardiovascular risk assessment, sleep duration, ejection fraction, glycemia level, Borg scale, etc.). Please specify the variables included in the data collection and extraction section, particularly those related to the tables.
- The authors repeat phrases in the Methods section (e.g., “We included patients admitted to rehabilitation in 2023”).
- The statistical methods section is insufficiently detailed and should be expanded.
Discussion:
- Please rewrite the opening sentences of the Discussion section for clarity. Additionally, the authors use abbreviations (e.g., CMR factors) in their conclusions that were not defined earlier in the study.
Round 2
Reviewer 1 Report
Comments and Suggestions for Authors
I consider that all my comments have been properly addressed and that the manuscript can be published in its current form.